# Feasibility of Child-Resistant and Senior-Friendly Press-Through Packages: Potential of Different Materials

**DOI:** 10.3390/pharmaceutics15030890

**Published:** 2023-03-09

**Authors:** Kiyomi Sadamoto, Hiroyuki Ura, Mikio Murata, Masaho Hayashi, Kiyoshi Kubota

**Affiliations:** 1Department of Clinical Pharmacy, Shonan University of Medical Science, 16-48 Kamishinano Totsuka, Yokohama 244-0806, Japan; 2MECSION, General Incorporated Association, 242-3 Guzo Hiratsuka, Hiratsuka 254-0906, Japan; 3Sadamoto Clinic, 821-10 Ninomiya, Naka, Yokohama 259-0123, Japan

**Keywords:** press-through package, older adults, children, child-resistant and senior-friendly, soft aluminum material, rheumatoid arthritis

## Abstract

Press-through packaging (PTP) is the most common type of drug packaging in Japan, and a production procedure for PTP has been established at an acceptable cost. However, unknown problems and new needs with regard to safety among users of various age-groups still need to be examined. Considering accident reports involving children and older adults, the safety and quality of PTP and new forms of PTP, such as child-resistant and senior-friendly (CRSF) packaging, should be evaluated. We conducted an ergonomic study on children and older adults to compare types of commonly used PTP and new varieties of PTP. Opening tests were attempted by children and older adults using a common type of PTP (Type A) and child-resistant (CR) PTP (Types B1 and B2) made from soft aluminum foil. The same opening test was conducted on older patients with rheumatoid arthritis (RA). The results showed that CR PTP was difficult for children to open: only 1 out of 18 children could open Type B1. On the other hand, all eight of the older adults could open Type B1, and eight patients with RA could easily open Types B1 and B2. These findings suggest that the quality of CRSF PTP can be improved with the use of new materials.

## 1. Introduction

Press-through packaging (PTP) is the most popular type of drug packaging for prescription and over-the-counter (OTC) drugs in Japan [1,2], meaning that PTP is used by not only patients with chronic diseases who use prescription drugs frequently but also individuals who occasionally use OTC medications (i.e., all generations). Of course, safety is the priority, but the manner of use depends on the individual user, and various difficulties and dangers remain unknown [3,4]. For example, accidents have been reported among older adults who cut PTP sheets to place drugs in a personal adherence box (a pill box the user keeps their important drugs in for ease of remembering to take their medication), where they sometimes inadvertently ingest portions of the cut PTP sheets along with their medication. These accidents are typically discovered during gastric endoscopy, suggesting that these dangerous accidents are not discovered in a timely manner [5]. In addition, older adults sometimes find it difficult to open PTP [6,7], and small children are sometimes involved in accidents in which they remove family members’ drugs from PTP and swallow them, which can lead to life-threatening consequences. The number of cases of these and other types of incidents involving medications is increasing [8,9]. All of these factors may affect actual adherence among patients [10,11]. Therefore, both situation-dependent drug use among older adults and related activities involving children need to be evaluated. Moreover, to improve the cost-effectiveness of drug distribution in medical care, PTP could potentially be capable of maintaining pills (more than 80% of oral drugs are in pill form) and capsules in a reasonable condition for the needed duration of time at an affordable cost before distribution [12,13]. However, to our knowledge, the quality of PTP has not been studied in detail. Designing PTP with improved quality and safety for both children and older adults could provide substantial benefits for medical use [14,15,16]. In the EU and US, guidelines have been devised for child-resistant and senior-friendly (CRSF) drug packages, but such guidelines are still lacking in Japan [17,18]. Given this background, the present study aimed to investigate the feasibility of CRSF PTP in Japan [19,20]. To evaluate the ease of opening of newly designed PTP, we conducted an ergonomic study involving children, older adults, and patients with rheumatoid arthritis (RA) to compare common types of PTP (Type A) with new types of PTP with a CRSF function (Types B1 and B2) in terms of ergonomics and materials (Figure 1).

## 2. Materials and Methods

To evaluate the quality of PTP, which is most commonly used for drug packages in Japan designed with a CRSF function, we conducted an ergonomic study and compared popular types of PTP with new ones among both children and older adults. Practical tests were performed with children, older adults, and patients with RA using Types A (normal commonly used PTP) and B (CRSF PTP B1 and B2; Type B1 is thicker than B2, i.e., Type B1 requires more force to push through and break), which were newly designed sample PTP made of soft aluminum foil, the structure and quality profiles of which are shown in Figure 1 and Figure 2, respectively. There were no relationship data between these profiles and PTP quality (e.g., ease of opening); therefore, we tried to consider this in our study.

Under the CRSF test (EN 14,375 EU standard; Table 1), children were asked to open 10 pockets of a PTP sheet. A timekeeper recorded the number of tablets they could open and remove from the PTP within 300 s. All of these children’s tests were done in the same room under the same conditions. They participated in the same opening tests. Eight older adults and eight patients with RA performed a similar test. The older group rated the ease of opening using a score (Figure 3), while the parents of the children in the children’s group answered a questionnaire on the performance of the children in relation to drugs in general at home. To examine the social problem related to drug taking in everyday life, we administered the questionnaire for parents of children. The children and patients with RA attempted to open Types B1 and B2. The performance for each type is summarized in Table 2. There was an interval for subjects who opened twice (3–4 min). Throughout all of the tests, there was no information on the quality of PTP, as the test was blinded. For analysis of patients with RA, pinch power was measured.

In this study, we performed paired t-tests or one-way analysis of variance (ANOVA) to determine the parametric difference between two sample groups or among three groups, respectively. The data obtained were analyzed using GraphPad Prism version 9.5.1 (GraphPad Software, San Diego, CA, USA). The level of statistical significance was set at *p* < 0.05.

## 3. Results

Figure 1 shows the PTP and tablets used in this study. For all types, the size of the tablets was nearly identical and the space for the tablets in the pockets was sufficient. Figure 2 shows the load profiles of Types A and B1. A clear difference was found between the two types. The load required to break the aluminum seal (N) was significantly higher for Type B1 than that for Type A (*p* < 0.01, t = 4.743, 49.3 ± 5.7, 30.5 ± 3.0, respectively).

Table 2 shows the characteristics of the three aluminum samples. The force (N) required to push the tablet out was measured using FORCE TESTER (MCT-1150/2150 A&D Co., Ltd., Tokyo, Japan). The results showed that Types B1 and B2 required 49.3 and 44.3 N, respectively, and Type A required 30.5 N. Types B1 and B2 both required more than 1.5 times the force compared with Type A. Types B1 and B2 also exhibited clearly higher elongation (12.7% and 12.5%, respectively) than Type A (2.25%) before breaking.

On the other hand, tensile strength and yield stress were clearly lower for Types B1 and B2 than for Type A. Pulling strength for Type A was about 2.5 times higher than that of Type B1. Yield stress for Type A (160 N/mm^2^) was four times higher than that of Types B1 and B2 (40.0 N/mm^2^), and the bursting strength of Type B1 (285 kPa) was almost twice as high as that of Type A (155 kPa).

Table 3a shows the results of the children’s tests. All 18 children (mean age, 51,1 ± 4.8 months; age range, 44–60 months) could open Type A within the given time; however, only one child could open Type B1 within the same period of time. Among the other 17 children, 15 stopped the opening trial by themselves and two tried for all 300 s before running out of time. The CR ratio was 94%. The mean number of opened tablets was significantly lower for Type B1 than for Type A (*p* < 0.0001, t = 12.45). Additionally, there was a statistically significant difference between the times elapsed before the pill was opened between Type A and Type B1 (*p* < 0.0001, t = 12.19). Table 3b shows the results for Type B2. As the aluminum of Type B2 was thinner than that of Type B1, four of the 18 children could open the PTP within 300 s. The CR ratio was 78%. The mean number of opened tablets was significantly lower for Type B2 than for Type A (*p* < 0.0001, t = 5.803). Additionally, there was a statistically significant difference between the times elapsed before the pill was opened between Type A and Type B2 (*p* < 0.0001, t = 5.406).

Table 4 shows the results for the older adults (mean age 70.1 ± 7.1 years, range 61–78 years). All participants could open Type B1. Seven of the participants removed all 10 tablets within the given time of 60 s. For the CRSF test to be successful, a subject only had to be able to open one pocket and remove one tablet within 60 s. The ease of opening score was 3.4 (range 3–4). No clear differences were found compared with conventional PTP.

There were three groups of study results. None of the participants had mental disabilities. RA participants had hand deformities depending on their stage of disease, and their difficulty in handling things depended on their disease class (Table 5). Table 5 shows the results for the patients with RA (mean age 65.4 ± 8.7 years, range 54–78 years). In total, six subjects were stage IV (Steinbrocker), two were stage III, and all had some deformity and dysfunction in their fingers. All subjects were female volunteer patients with RA (mean age 65.4 ± 8.7 years, mean pinch force 27.3 ± 9.2 N, range 13–40 N). In the opening trial, Type B1 was evaluated first, considering the participants’ finger deformities/dysfunction. All patients could open Type B1. Next, we tried Type B2. Again, all patients could open the PTP and remove one tablet within the given time of 60 s. In fact, five of the patients were able to remove one tablet within 5 s. There were no statistically significant differences between the times required to dispense a tablet from the packaging among Type A, Type B1, and Type B2 (*p* = 0.63, F = 0.36). Observations of the methods of opening and removing the tablet revealed that seven patients with RA used their fingers and one used her fingernail to break the aluminum, and then used two fingers to push the tablet out.

Figure 4a shows a chart detailing the relationship between the number of packages opened and not opened in the children’s test. This chart shows the acceptable zone for CR function. Type B1 is located in the acceptable zone, whereas Type A is not. Figure 4b shows a chart for Type B2, which was above the acceptable zone. The CR ratio for Type B2 was 78%, slightly less than the required CR package quality (>80%).

Figure 5 shows the results of the questionnaire administered to the children’s parents. Sixteen of the parents observed that their children were interested in medicine for adults, and 14 that their children had actually physically picked up medication packages in their home. Three reported that their child had opened PTP for adults, and that one had actually ingested their mother’s medicine (5.6%). In the free comment section of the questionnaire, the parents described difficulties in giving necessary medicine to their children, as well as potentially dangerous accidents involving children playing with inhaler devices and mistaking medicine meant for adults for sweets (Table 6).

## 4. Discussion

Although PTP is frequently used in everyday life, it is difficult to compare differences in quality among various types of PTP. However, differences in quality affect usability, particularly the ease of opening PTP. Whether someone finds it difficult to open PTP is something that is not readily known to others. The fact that some children misuse PTP is also a problem. Looking at the reality of drug therapy in medical care, the numbers and variety of pills and capsules, including new and generic drugs, are increasing; thus, more attention is needed with regard to the correct use of all these drugs. As adherence reflects the effectiveness of drugs, good adherence must be maintained considering various factors. Drug packages must consist of easily understandable instructions, colors, and designs and be easy to handle, especially for older patients, as drugs are one of the most important elements of their daily life. Given this background, the EU and US have devised rules for senior-friendly drug packages, as well as CRSF test guidelines. However, to date, no such rules or guidelines have been devised for CRSF drug packages in Japan. As almost all prescription and OTC drugs in Japan are distributed in PTP, reasonably designed PTP with CRSF functionality needs to be proposed. It is necessary to examine human ergonomic studies with several user groups, including children. It is also understood that human ergonomic studies need precise observations, so we performed the test only among older adults (who regularly use the drug in outpatient clinics), patients with RA (who are a member of the RA association and use drugs regularly), and children (who belong to a private kindergarten in a middle-class residential area) who understood the instructions given regarding the study and cooperated with the investigation. These subjects are representative of ordinary Japanese people of the same age, and so we adopted them as our study subjects. Since the same examiner observed all performances (which was an advantaged compared with large-scale examinations), the results were highly reliable for evaluation. Therefore, in the present study, we attempted to evaluate PTP made from soft aluminum as a candidate for CRSF PTP. We observed differences in the material properties between soft aluminum and the type of aluminum commonly used, resulting in our decision to use soft aluminum for the CRSF PTP [21]. From the analysis of the material, clear differences in the aluminum’s material performance were found in terms of stroke and load (Figure 2). The stroke of Type B1 is more than 1.3 times longer than that of Type A, and the load necessary to break the aluminum is about 1.7 times higher. The main design of the PTP (e.g., pocket shape, size) was similar for Types A, B1, and B2 (Figure 1), so the differences in material properties were attributed to the nature of the aluminum. In fact, the pushing force required to break the packaging for Type A was lower than that for Types B1 and B2; however, for all other attributes, including elongation, tensile strength, yield stress, and bursting stress, Type A showed higher values than Types B1 and B2 (Table 2). In particular, the property of elongation reflected the consistency of the material. This means that material change leads to enormous effectiveness for making appropriate CRSF packages. Furthermore, this change does not require major production procedure changes. This benefit could be effective in a practical proposal for CRSF PTP. Considering this background, we conducted an ergonomic study to test the openability of individual PTP under the EU CRSF guidelines (Table 1 and Table 2).

Table 3a shows the results of the opening test among the first group of children. For Type A, all 18 children were able to open the PTP and remove the pills within 300 s, and 13 could do so within 100 s. On the other hand, only one child was able to open Type B1 within 300 s. Among the 17 children who could not open Type B1, 15 stopped the opening trial by themselves and two tried for all 300 s, but ran out of time. In the second group of children, all could open Type A, and four could open Type B2. The CR ratio was 78%. Because Type B2 is thinner than Type B1, it was easier to open. Thus, Type B1 could be reasonable for use as a CRSF PTP. Objective observations revealed that every child struggled to open Type B1, which they found to be different from Type A with regard to the difficulty of breaking it open. The nature of soft aluminum, which requires a long stroke to break, increases the difficulty experienced by children of opening packages made from it. Although we did not predict such a difficulty, the results of the children’s test clearly showed the intended functionality of child-resistant packaging. This means that changing the material is effective for improving the safety of such packaging around children. The only child who could open Type B1 was the same child who opened Type A in 40 s, which was the shortest time among all 18 children. Overall, most of the children did not have sufficient hand size or could not exert enough force to break the soft aluminum packages. Since most of the PTP we usually use are of medium size, like the one used in this study, the results may be appropriate for various drugs. However, further investigation using differently sized PTP is needed to make the results obtained in this study more universal.

Table 4 shows the results of the test conducted on the older adults (both male and female) who had no problem opening Type B1. Seven older adults were able to open the PTP within 60 s, and one took 84 s (she took extra time when starting). As a test was judged to be successful upon opening and removing one tablet within 1 min, all subjects were considered to have succeeded. In addition, the ease-of-opening score among these subjects was quite reasonable, as they evaluated Type B1 as being easy to use. The fact that these subjects, who use drugs frequently, were able to open Type B1 clearly indicates that the PTP is SF (i.e., soft aluminum packaging is easy for older adults to break open) [22,23].

Table 5 shows the results for patients with RA. The mean grip strength and pinch force of these patients are lower than those of adults of the same age, and so it is usually difficult for such patients to push and pinch things. In most opening tests using the EU standard, the authors provide information regarding the age, sex, and number of subjects in the test. In the real world, however, there are people with various handling disabilities, and so studying patients with RA is interesting and important, and the results give more practical potentiality. To evaluate the ease of opening the soft aluminum PTP, Types B1 and B2 were used. The results indicated that all eight patients with RA were able to push and break Types B1 and B2. In the trial for Type B1, all patients could push and break the PTP within 8 s. In the trial for Type B2, seven of the patients could push and break the PTP within 6 s, and the other within 18 s. These results demonstrate that even patients with some deformity and dysfunction in their fingers were able to use the newly designed PTP. The force required to break the packaging for Types B1 and B2 was 51 N and 30 N, respectively; however, patients with RA noted that they had not noticed a substantial difference between the PTP they use daily and the newly designed PTP. Therefore, even patients who have some deformity and dysfunction in their fingers can use PTP made from soft aluminum [24,25].

Figure 4a shows the chart, referring to EN14375, regarding the results of the sequential tests among children. The chart demonstrates the possibility of using the tested types of PTP in CR packages objectively. Type B1 was in a CR acceptable zone, whereas Type A was not. Figure 4b shows that Type B2 was in the rejection zone, so Type B1 appears to be the most reasonable choice for CRSF PTP. These findings correspond to the results of the human ergonomic test, suggesting that Type B1 is suitable as a CR package.

Figure 5 shows the questionnaire results among the children’s parents. In total, 12 of the parents had witnessed children taking medicine meant for their parent, which seems to reflect Japanese statistics indicating an increasing number of accidents involving accidental drug ingestion among children [8,9]. The free comments reporting that children mistook medicine for sweets also suggest the need for CR packaging.

The soft aluminum PTP used in this study has an identical appearance to ordinary PTP, which means that the manufacturing process, structure, pill size, and pocket space are the same as those for Type A. Only the material was different, but the results regarding the ease of opening among children were substantially different. Only one child was able to open Type B1, which indicated the quality of the CR packaging. On the other hand, the older adults and patients with RA were able to open the PTP and remove the contents, which indicated that changing the material resulted in CRSF functionality. In practice, there appears to be no need to change the manufacturing process, and there may be no obvious difference in production costs, indicating the potential for successful practical application. In addition, the lack of changes to the appearance of the PTP would also help avoid confusion among older patients.

Regarding pharmaceutical production and drug consumption, safety is always the priority; however, appropriate production costs are important from the perspective of sustainability. PTP made from different materials could help overcome the difficult challenge of creating CRSF PTP.

## 5. Conclusions

Worldwide, PTP is one of the most popular types of drug packaging, so it plays an important role in drug safety and distribution for users. Particularly in Japan, PTP is used for almost all prescription and OTC drugs. Given its ubiquity, some changes are needed to help prevent accidents involving medications among children. Newly designed PTP that is easier to use for older adults and patients with RA, among others, is needed; therefore, the proposition of CRSF PTP needs to be considered. In this study, we evaluated the usability of a new type of PTP made from soft aluminum. We observed differences in the material properties and usability among children. Because the difference in the material properties only affected usability among children, we were able to demonstrate the possible utility of CRSF PTP. Simple innovations in PTP materials could lead to the implementation of safer, more effective, and sustainable drug delivery methods for patients of all ages.

## Figures and Tables

**Figure 1 pharmaceutics-15-00890-f001:**
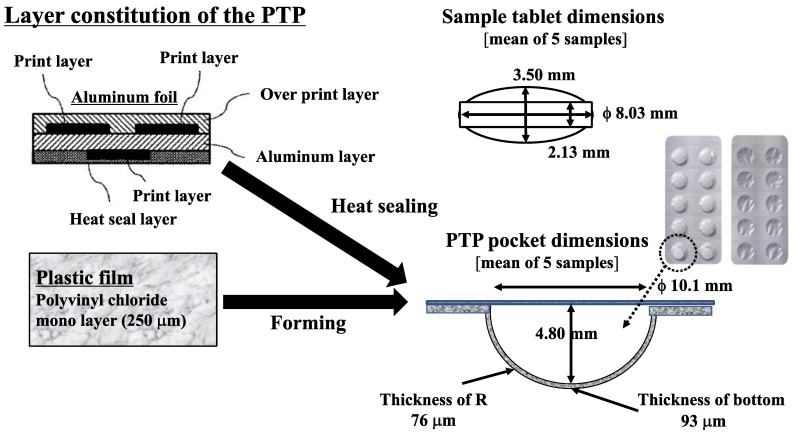
PTP materials and size of tablets and PTP pockets.

**Figure 2 pharmaceutics-15-00890-f002:**
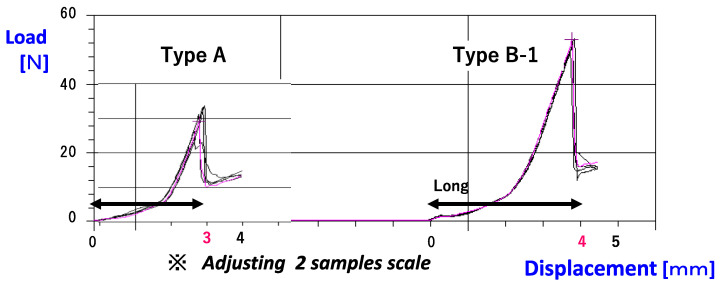
Load profile of samples A and B1. This profile was measured by a FORCE TESTER MCT-1150/2150 (φ5-mm cylinder, 10 mm/min, n = 5). Sample Type B1 requires a longer stroke before the aluminum breaks than Type A does.

**Figure 3 pharmaceutics-15-00890-f003:**
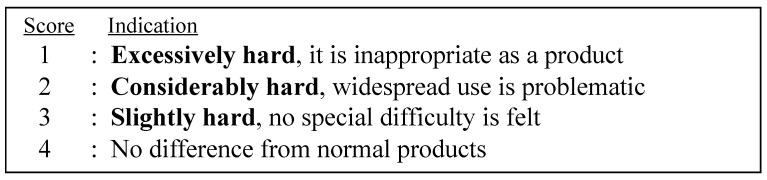
Ease of opening of sample Type B1. The score is the subjective evaluation after the CR adult test in the group of older adults.

**Figure 4 pharmaceutics-15-00890-f004:**
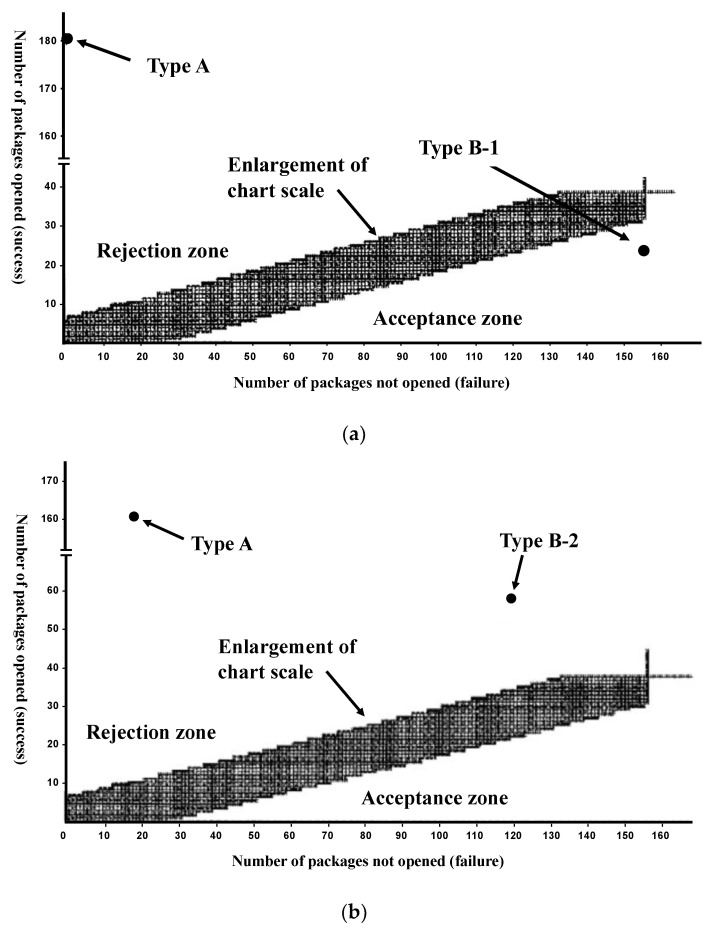
(**a**) Child Test-1. Type B1 plots on the chart are for the sequential tests among children. (**b**) Child Test-2. Type B2 plots on the chart are for the sequential tests among children.

**Figure 5 pharmaceutics-15-00890-f005:**
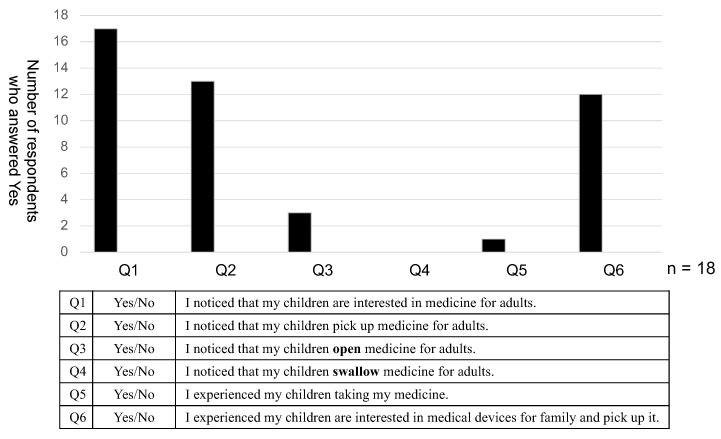
Questionnaire results for the parents of children and free comments.

**Table 1 pharmaceutics-15-00890-t001:** EN 14,375 child and adult test (for non-reclosable packaging).

**1. Child Test (for the Type A Test, the opening method is explained, so it is considered to take 10 minutes)**
Individual tests shall be considered a failure in relation to unit, strip, or blister packages if the child accesses more than 8 unit doses from the packaging provided within 10 min.
5.1 Principle: Child test (child aged 52–60 months is considered to be 51 months of age)
Type approval for non-reclosable child-resistant packaging is obtained by a sequential test method or full panel test for children. A test group of up to 200 children aged 42 to 51 months is divided into pairs. Each child is given a number of non-reclosable packages to be opened by whatever means they wish to use. If a child fails to gain access within 5 min, the method of opening is demonstrated by the supervisor and the child is given a further 5 min to open the packages. The results are recorded sequentially, as obtained. The package is deemed child-resistant if the trial results on the test charts pass into the acceptable zone [Figure 4 Chart] or if at least 80% of the children are unable to access 8 unit doses within 10 min and at least 85% of the children are unable to access more than 8 unit doses within the first 5 min.
5.3.2.1 Composition of the child test group
The test group shall comprise no more than 200 children aged 42–51 months with approximately equal numbers of girls and boys. As far as possible, there shall be an even distribution of ages and sexes within the panel. The children shall be selected at random and shall have no apparent physical or mental disability that might affect manual dexterity. They shall not have taken part in more than one previous test, and in that test, a packaging of a different type and design shall have been used. If a child is used for more than one test, there shall be at least 4 weeks between tests. Parental or guardian consent shall be obtained before the child is used as a part of the test group. Any children having been involved in a reported poisoning accident shall be excluded from this test.
**2. Adult test**
When tested in accordance with 5.3.3.2, at least 90% of the adults shall be able to access at least 1 unit dose within the 1 min test period, without demonstration.
5.1 Principle: Adult test (we prepared healthy older adults and patients with RA in this adult test panel)
The package’s accessibility by a test group of 100 adults is also assessed. Each adult is given a non-reclosable package, any associated opening tools, and written instructions, and is allowed 5 min to familiarize themselves with the packaging. The number of adults opening the package within a 1 min test period is recorded. The package is deemed to comply with the requirements of this standard if at least 90% of the adults are able to access at least 1 unit dose in 1 min.
5.3.3.1 Composition of the adult test group
The test group shall comprise 100 participants. These shall be selected using a screening procedure in which potential participants shall be asked the following questions: “Are you professionally concerned with the design, manufacture, or use of child-resistant packaging?” and “Have you taken part in more than one previous child-resistant packaging test within 6 months?” Only those participants responding with negative answers shall be selected. Persons with obvious physical disabilities that might affect manual dexterity shall not be approached, and those unable to understand the written opening instructions shall be excluded. The 100 participants shall be randomly selected from individuals between the ages of 50 and 70 years.

**Table 2 pharmaceutics-15-00890-t002:** Performance of the aluminum sample.

	Conventional Aluminum Foil (Type A)	Prototype Aluminum Foil for Child Resistance		Reference Data ^¶^
		Type B1	Type B2	
Type of aluminum foil	hard	soft	soft	hard
Grade of aluminum foil *	IN30	8079	8079	IN30
Thickness [μm]	20	30	25	30
Force required to break [N] ^†^	30.5	49.3	44.3	43.2
Elongation [%] ^‡^	2.25	12.7	12.5	9.10
Tensile strength [N/mm2] ^§^	185	77.0	79.5	64.0
Yield stress [N/mm2] ^§^	160	40.0	40.0	31.5
Bursting strength [kPa] ^||^	155	285	230	225

* Alloy number by JIS H4160-1994 “Aluminum and aluminum alloy foils”. ^†^ FORCE TESTER MCT-1150/2150 [A&D Co., Ltd., Tokyo, Japan] (f 5mm cylinder, 10 mm/min). ^‡^ JIS Z 2241 METALLIC MATERIALS – TENSILE TESTING. § JIS L 1096 A (strip) method. ^||^ JIS K 6404-11 (ISO 3303) Mullen tester. ^¶^ Reference data were collected by SANKO Alumi Inc. (Saitama, Japan).

**Table 3 pharmaceutics-15-00890-t003:** (**a**) CR Child Test-1 (Type B1). (**b**) CR Child Test-2 (Type B2).

(**a**)						
	**Panel**		**Type A (Hard AL 20 μm)**		**Type B1 (Soft AL 30 μm)**	
	**Sex**	**Age (months)**	**Opening Tablets**	**Time (s)**	**Opening Tablets**	**Time (s)**
1	FM	60	10	238	7	300
2	FM	50	10	110	0	300
3	M	59	10	83	0	300
4	M	59	10	83	0	300
5	FM	57	10	40	10	138
6	M	52	10	80	0	300
7	FM	51	10	77	0	300
8	M	56	10	67	0	300
9	FM	53	10	99	0	300
10	M	51	10	80	0	300
11	FM	58	10	275	0	300
12	M	50	10	153	0	300
13	M	59	10	65	2	300
14	FM	56	10	44	5	300
15	M	52	10	56	0	300
16	M	52	10	84	0	300
17	M	56	10	68	2	300
18	M	59	10	144	0	300
Mean of opening tablets			10.0		1.4	
CR ratio			0% [0/18]		94% [17/18]	
(**b**)						
	**Panel**		**Type A (Hard AL 20 μm)**		**Type B2 (Soft AL 25 μm)**	
	**Sex**	**Age (months)**	**Opening Tablets**	**Time (s)**	**Opening Tablets**	**Time (s)**
1	FM	49	7	300	0	300
2	FM	45	10	114	6	300
3	M	46	5	300	0	300
4	M	44	10	239	0	300
5	FM	50	10	150	10	140
6	FM	51	10	75	5	300
7	M	48	10	106	0	300
8	FM	44	10	51	0	300
9	M	45	10	41	10	192
10	M	51	10	62	10	247
11	FM	50	10	176	0	300
12	M	51	10	87	1	300
13	FM	47	10	43	10	72
14	M	50	10	109	1	300
15	M	50	10	92	0	300
16	M	46	10	80	7	300
17	M	44	7	300	0	300
18	FM	47	1	300	0	300
Mean of Opening Tablets			8.9		3.3	
CR ratio			22% [4/18]		78% [14/18]	

**Table 4 pharmaceutics-15-00890-t004:** CR Adult Test-1 on healthy older adults. The senior-friendly (SF) ratio is the ratio of the panel able to open the packaging within 1 min.

	Healthy Older Adult Panel		CR Adult Test Type B1		
	Sex	Age (years)	1 min Test [Time]	Reference: Time Required for 10 Tablets of Opening	Score of Opening Status
1	M	78	○ [< 5″]	<60″	4
2	FM	78	○ [< 5″]	<60″	3
3	M	76	○ [< 5″]	84″	4
4	FM	74	○ [< 5″]	<60″	4
5	FM	69	○ [< 5″]	<60″	3
6	M	63	○ [< 5″]	<60″	3
7	FM	62	○ [< 5″]	<60″	3
8	M	61	○ [< 5″]	<60″	3
		Mean: 70	SF Ratio: 100% [8/8]	—	Mean: 3.4

**Table 5 pharmaceutics-15-00890-t005:** CR Adult Test-2 on patients with RA.

	RA Patient Panel				CR Adult Test		
	Sex	Age	Pinch Force [N]	RA Stage/Class	1 min Test [Time]		
					Type A (Hard AL 20 μm)	Type B1 (Soft AL 30 μm)	Type B2 (Soft AL 25 μm)
1	FM	78	13	Ⅳ/Ⅱ	○ [5″]	○ [4″]	○ [5″]
2	FM	61	32	Ⅲ/Ⅱ	○ [3″]	○ [2″]	○ [4″]
3	FM	68	40	Ⅲ/Ⅱ	○ [3″]	○ [6″]	○ [8″]
4	FM	68	20	Ⅲ/Ⅱ	○ [4″]	○ [18″]	○ [6″]
5	FM	55	35	Ⅲ/Ⅱ	○ [5″]	○ [5″]	○ [2″]
6	FM	75	19	Ⅲ/Ⅱ	○ [8″]	○ [2″]	○ [4″]
7	FM	54	27	Ⅲ/Ⅱ	○ [2″]	○ [2″]	○ [2″]
8	FM	64	32	Ⅳ/Ⅱ	○ [2″]	○ [3″]	○ [2″]
Mean		63	27	—	SF Ratio: 100% [8/8]	SF Ratio: 100% [8/8]	SF Ratio: 100% [8/8]

**Table 6 pharmaceutics-15-00890-t006:** Free comments about children.

1	It is difficult for my children to take bitter medicine.
2	It is difficult for my children to take bitter medicine and use eye drops.
3	It is difficult for my children to take bitter medicine. They vomit, so it is difficult to give the appropriate amount of medicine.
4	My child sometimes plays with his brother’s inhaler.
5	My child took medicine because he mistook it for sweets. On the other hand, he did not take medicine that he needed to take.

## Data Availability

The data presented in this study are available on request from the corresponding author.

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
