# Peer review of "Feasibility of Child-Resistant and Senior-Friendly Press-Through Packages: Potential of Different Materials"

_pharmaceutics, 2023, doi:10.3390/pharmaceutics15030890_

Round 1

Reviewer 1 Report

This is a well written article on an important aspect addressing the safety of medicines. However the manuscript needs further work. 

Major:

1) All methods used should be clearly described in the method section. This would inlude e.g. the development of the questionnaire, measurement of pinch grip, selected tablet type with key characteristcs incluiding tablet hardness etc.

2) The results that are gathered with the tests indicated in the method section should only be described in the results section, but without any further interpretation.

3) The justification of the method seclection plus the interpretation of the results plus an evaluation of results versus those of other authors plus the strenght and limitations of this study should be desrirbed in the the discussion section.

Minor:

4) The small number of children and senior needs further justification. Their age and gender should be indicated in the results section

5) The generalizability of the study findings towards smaller/larger and more/less hard tablets should be discussed.    

Author Response

According to the reviewers’ comments, or “English language and style are fine/minor spell check required.”, We have undergone native English checks and English editing to polish our manuscripts.

[Comment 1]: All methods used should be clearly described in the method section. This would inlude e.g. the development of the questionnaire, measurement of pinch grip, selected tablet type with key characteristcs incluiding tablet hardness etc.

[Response 1]: Thank you for the comment. According to the reviewer’s comment, we added the sentences in the Material and Methods, as follows:

  • Page 2, line 73: “To examine the social problem related to drug taking in everyday life, we administered the questionnaire for parents of children.”
  • Page 2, line 75: “There was an interval time for subjects who opened twice, and it was more than three minutes (3-4 min). Throughout all of the tests, there was no information on the quality of PTP, as the test was blinded. For analysis in the human ergonomic study with patients with RA, pinch power was measured. This could represent some degree of difficulty among them.”

[Comment 2]: The results that are gathered with the tests indicated in the method section should only be described in the results section, but without any further interpretation.

[Response 2]: Thank you for the important point. We moved the series of words from the Materials and Methods to the Results, as below:

  • From line 66 to line: 94: “(mean age, 51,1 ± 4.8 months; age range, 44–60 months)”
  • From line 69 to line 101: “(mean age, 70.1 ± 7.1 years; age range, 61–78 years)”
  • From line 70 to line 106: “(mean age, 65.4 ± 8.7 years; age range, 54–78 years)”

[Comment 3]: The justification of the method seclection plus the interpretation of the results plus an evaluation of results versus those of other authors plus the strenght and limitations of this study should be desrirbed in the the discussion section.

[Response 3]: Thank you for the suggestion of justification. We added the sentences in the Discussion. (page 4, line 146), “It is necessary to examine human ergonomic studies with several user groups, including children. It is also understood that human ergonomic studies need precise observations, so we performed the test only among older adults (who regularly use the drug in outpatient clinics), patients with RA (who are a member of the RA association and use drugs regularly) and children (who belong to a private kindergarten in a middle-class residential area) who understood the instructions given regarding the study and cooperated with the investigation. These subjects are representative of ordinary Japanese people of the same age, and so we adopted them as our study subject. Since the same examiner observed all performances (which was an advantaged compared with large-scale examinations), the results were highly reliable for evaluation.”

Additionally, we changed the following sentence in the Discussion (page 5, line 201), “Figure 4-1 shows a chart regarding the results of the sequential tests among children.” to “Figure 4-1 shows the chart, referring to EN14375, regarding the results of the sequential tests among children.”

[Comment 4]: The small number of children and senior needs further justification. Their age and gender should be indicated in the results section

[Response 4]: Thank you for the suggestion. we added the sentences in the Discussion (page 4, line 146, please see Response 3).

[Comment 5]: The generalizability of the study findings towards smaller/larger and more/less hard tablets should be discussed.

[Response 5]: Thank you for the useful suggestion. We added the sentences in the Discussion, as follows:

  • Page 4, line 159: “This means that material change leads to enormous effectiveness for making appropriate CRSF packages. Furthermore, this change does not require major production procedure changes. This benefit could be effective in the practical proposal for CRSF PTP.”
  • Page 4, line 178: “Since most of the PTP we usually use are medium size, like the one used in this study, the results may be appropriate for various drugs. However, further investigation using different-sized PTP is needed to make the results obtained in this study more universal.”

Again, thank you for giving us the opportunity to strengthen our manuscript with your valuable comments and queries. We have worked hard to incorporate your feedback and hope that these revisions persuade you to accept our submission.

Sincerely,

Kiyomi Sadamoto, M.D., Ph.D.

Professor

Department of Clinical Pharmacy, Shonan University of Medical Science

Reviewer 2 Report

The authors conducted an experimental investigation to compare the ease-of-opening of a newly designed Press-through package (PTP) with that of a common type of PTP. By conducting opening tests involving children, older adults, and patients with rheumatoid arthritis, the authors showed that the newly designed PTP was less likely to be opened by children but easily opened by older adults and patients with rheumatoid arthritis. The findings are interesting. There are some comments.

  1. Materials and Methods: Children, older adults, and patients with RA were recruited in the study for the opening test. However, it is unclear how these individuals were sampled and recruited. The following information is lacking and would be informative: the eligibility criteria, the number of individuals assessed for eligibility, the number of individuals noted as ineligible (in total and for each eligibility criterion), and the number of individuals who were confirmed eligible and recruited. 
  2. Materials and Methods: From the description, it is unclear whether a child, older adult, or patient with RA took part in more than one opening test. If they took part in more than one test, the time interval between tests is unclear. Also, it is unclear whether blinding was performed.
  3. Materials and Methods: Differences (for instance, load profiles of PTP-A and PTP-B, ease-of-opening score) were compared in this study. It is advised to apply formal statistical tests in examining the differences.
  4. Results:  Please present the characteristics (for instance, age, sex, physical or mental disability) of the recruited children, older adults, and patients with RA.
  5. Table 2: It is advised to conduct statistical tests to test the differences in the performance of the aluminum samples and present the results.
  6. Figure 2: It is advised to conduct statistical tests to test the differences in the load profiles of the aluminum samples and present the results.
  7. Table 3-1 and Table 3-2: It is advised to conduct statistical tests to test the differences and present the results.
  8. Table 4 and Table 5: Please present the results of the CR test of type A. Also, it is advised to conduct statistical tests to test the differences and show the results.
  9. Figure 4: Please recheck the correctness of the labels of the X-axis (number of packages not opened?) and the Y-axis (number of packages opened?).
  10. Discussion: According to the presented EN 14375 CHILD AND ADULTS TEST (table 1), “persons with obvious physical disabilities that might affect manual dexterity shall not be approached, -.” However, patients with RA and deformity and dysfunction in their fingers were included in this study (Line 131 on Page 6). A discussion regarding the potential implications is suggested.

Author Response

According to the reviewers’ comments, or “English language and style are fine/minor spell check required.”, We have undergone native English checks and English editing to polish our manuscripts.

[Comment 1]: Materials and Methods: Children, older adults, and patients with RA were recruited in the study for the opening test. However, it is unclear how these individuals were sampled and recruited. The following information is lacking and would be informative: the eligibility criteria, the number of individuals assessed for eligibility, the number of individuals noted as ineligible (in total and for each eligibility criterion), and the number of individuals who were confirmed eligible and recruited.

[Response 1]: Thank you for the important points. We added the following sentences in the Discussion (page 4, line 146) according to the suggestion of the reviewer 1 and 2, “It is necessary to examine human ergonomic studies with several user groups, including children. It is also understood that human ergonomic studies need precise observations, so we performed the test only among older adults (who regularly use the drug in outpatient clinics), patients with RA (who are a member of the RA association and use drugs regularly) and children (who belong to a private kindergarten in a middle-class residential area) who understood the instructions given regarding the study and cooperated with the investigation. These subjects are representative of ordinary Japanese people of the same age, and so we adopted them as our study subject. Since the same examiner observed all performances (which was an advantaged compared with large-scale examinations), the results were highly reliable for evaluation.”

[Comment 2]: Materials and Methods: From the description, it is unclear whether a child, older adult, or patient with RA took part in more than one opening test. If they took part in more than one test, the time interval between tests is unclear. Also, it is unclear whether blinding was performed.

[Response 2]: Thank you for the precise observation. We changed the following sentence in the Materials and Methods (page 2, line 65) “To evaluate the quality of PTP designed with a CRSF function, …” to “To evaluate the quality of PTP, which is most commonly used for drug packages in Japan designed with a CRSF function, …”. Additionally, we added the information in the Materials and Methods, as below:

  • Page 2, line 69: “All of these children’s tests were done in the same room under the same conditions. They participated in the same opening tests”
  • Page 2, line 75: “There was an interval time for subjects who opened twice, and it was more than three minutes (3-4 min). Throughout all of the tests, there was no information on the quality of PTP, as the test was blinded. For analysis in the human ergonomic study with patients with RA, pinch power was measured. This could represent some degree of difficulty among them.”

[Comment 3]: Materials and Methods: Differences (for instance, load profiles of PTP-A and PTP-B, ease-of-opening score) were compared in this study. It is advised to apply formal statistical tests in examining the differences.

[Response 3]: Thank you for the advice. There is, unfortunately, no generalized tests for PTP quality difference (sample size of 2, Table 2). However, it is interesting to analyze relationship among items of Table 2 and quality of PTP. Further consideration will be given in the future. We added the sentence in the Materials and Methods (page 2, line 65), “There were no relationship data between these profiles and PTP quality (e.g., ease-of-opening); therefore, we tried to consider this in our study.”

[Comment 4]: Results: Please present the characteristics (for instance, age, sex, physical or mental disability) of the recruited children, older adults, and patients with RA.

[Response 4]: Thank you for the useful information. We added the following sentences in the Results (page 3, line 106), “There were three groups of human ergonomic study results. All of the participants had no mental disabilities. RA participants had hand deformities depending on their stage of disease, and their difficulty of handling things depended on their disease class (Table 5).”

Additionally, we moved the series of words from the Materials and Methods to the Results, as below:

  • From line 66 to line: 94: “(mean age, 51,1 ± 4.8 months; age range, 44–60 months)”
  • From line 69 to line 101: “(mean age, 70.1 ± 7.1 years; age range, 61–78 years)”
  • From line 70 to line 106: “(mean age, 65.4 ± 8.7 years; age range, 54–78 years)”

[Comment 5]: Table 2: It is advised to conduct statistical tests to test the differences in the performance of the aluminum samples and present the results.

[Response 5]: Thank you for the advice. The data in Table 2 are difficult to describe statistically with a sample size of 2. However, since the samples are industrially prepared, there should be little variation among each sample.

[Comment 6]: Figure 2: It is advised to conduct statistical tests to test the differences in the load profiles of the aluminum samples and present the results.

[Response 6]: Thank you for the advice. We have performed paired-t-test between Type A and Type B of the load to break the aluminum seal. our manuscript was revised as follows:

  • We changed the sentences according to the reviewer’s suggestion (Page 2, line 79): “A clear difference was found between the two types. The load required to break the aluminum seal (N) was significantly higher for Type B1 than that for Type A (P < 0.01, t = 4.743, 49.3 ± 5.7, 30.5 ± 3.0, respectively)
  • We added the following sentences in the Materials and Methods (page 2, line 75), “In this study, we performed paired t-tests or one-way analysis of variance (ANOVA) to determine the parametric difference between two sample groups or among three groups, respectively. The data obtained were analyzed using GraphPad Prism version 9.5.1 (GraphPad Software, San Diego, USA). The level of statistical significance was set at P < 0.05.

[Comment 7]: Table 3-1 and Table 3-2: It is advised to conduct statistical tests to test the differences and present the results.

[Response 7]: Thank you for the advice. We have performed paired-t-test between the groups of opening tablets and time to remove pills, and we added the sentences in the Results, as follows:

  • Page 3, line 97 in the Results of the previous manuscript: “The mean number of opened tablets was significantly lower for Type B-1 than for Type A (P < 0.0001, t = 12.45). Additionally, there was a statistically significant difference between the times elapsed before the pill was opened between Type A and Type B-1 (P < 0.0001, t = 12.19).”
  • Page 3, line 100 in the Results of the previous manuscript: “The mean number of opened tablets was significantly lower for Type B-2 than for Type A (P < 0.0001, t = 5.803). Additionally, there was a statistically significant difference between the times elapsed before the pill was opened between Type A and Type B-2 (P < 0.0001, t = 5.406).”

[Comment 8]: Table 4 and Table 5: Please present the results of the CR test of type A. Also, it is advised to conduct statistical tests to test the differences and show the results.

[Response 8]: Thank you for the advice. We added the results of the CR test of type A in Table 5. We also have performed the one-way analysis of variance among the groups according to the reviewer’s suggestion, and we added the following sentence in the Results (page 3, line 113), “There were no statistically significant differences between the times required to dispense a tablet from the packaging among Type A, Type B-1 and Type B-2 (P = 0.63, F = 0.36).”

[Comment 9]: Figure 4: Please recheck the correctness of the labels of the X-axis (number of packages not opened?) and the Y-axis (number of packages opened?).

[Response 9]: Thank you for the helpful suggestion. We had been typing incorrectly with respect to the X- and Y-axes. We corrected Figure 4-1 and 4-2 according to the reviewer’s suggestion (X-axis was not opened, Y-axis was opened). In addition, we have made changes to the X- and Y-axis notations in Figure 5 to clarify the meaning of the graphs.

[Comment 10]: Discussion: According to the presented EN 14375 CHILD AND ADULTS TEST (table 1), “persons with obvious physical disabilities that might affect manual dexterity shall not be approached, -.” However, patients with RA and deformity and dysfunction in their fingers were included in this study (Line 131 on Page 6). A discussion regarding the potential implications is suggested.

[Response 10]: Thank you for the important points. We added the following sentences in the Discussion (page 5, line 190), “In most opening tests using the EU standard, the authors provide information regarding the age, sex, and number of subjects in the test. In the real world, however, there are people with various handling disabilities, and so studying patients with RA is interesting and important, and the results give more practical potentiality.”

Again, thank you for giving us the opportunity to strengthen our manuscript with your valuable comments and queries. We have worked hard to incorporate your feedback and hope that these revisions persuade you to accept our submission.

Sincerely,

Kiyomi Sadamoto, M.D., Ph.D.

Professor

Department of Clinical Pharmacy, Shonan University of Medical Science